# SHOULD WE FORGET ABOUT CERTIFIED UNLEARNING? EVALUATING THE PITFALLS OF NOISY METHODS

## ABSTRACT

Removing the influence of certain training data points from trained models ("unlearning") is a critical need driven by data privacy regulations. While a straightforward way to achieve this "exactly" is to retrain from scratch on only permissible data (the "retain set"), that approach is computationally prohibitive. A promising alternative involves first training a model on the full dataset with differential privacy (DP) and then fine-tuning it, with or without noise, on only the retain set. This offers certifiable unlearning: while unlearning is approximate in this case, this method comes with theoretical guarantees on the quality of that approximation, building on the DP guarantees. Recent papers claim that this approach makes favourable tradeoffs relative to retraining: while DP-unlearning offers a weaker guarantee, and may degrade model utility, it is more efficient. However, the practical viability of this approach has not been rigorously assessed in realistic settings. We conduct a systematic evaluation across both vision and language tasks revealing that, contrary to prevailing claims, DP-unlearning methods fail to offer a compelling advantage over retraining from scratch, even after applying several improvements to maximize their potential, and even when allowing them to offer a weaker guarantee than what would be necessary in some practical scenarios. We identify two key failure modes explaining this result. First, if starting from a random initialization, DP guides models to suboptimal solutions from which they cannot easily escape, costing too much in terms of utility. On the other hand, starting the training from a pretrained model doesn't pay off either: simply "re-finetuning" that pretrained model is already quite fast, while also having the strongest unlearning guarantee. Overall, we failed to find a scenario where certified unlearning is worthwhile. This important negative result highlights the need to explore alternative techniques.

## 1 INTRODUCTION

Recent developments in data privacy regulations, including the European Union's General Data Protection Regulation (GDPR), the California Consumer Privacy Act (CCPA), and Canadian Consumer Privacy Protection Act (CPPA) have introduced legal notions like the "right to be forgotten," which have spurred the development of *machine unlearning* (Bourtoule et al., 2021; Nguyen et al., 2024; Cao & Yang, 2015). In this context, the goal of an unlearning algorithm is to efficiently remove the influence of a particular data sample $x \in D$ (or set of samples $D_F \subset D$), from a trained model $M$. In the ideal case, the resulting unlearned model should behave identically to a model trained from scratch on *only* the remaining data, or retain set, $D_R = D \setminus D_F$. While simply retraining the model on $D_R$ yields perfect unlearning, this method, known as retraining from scratch (or RFS), is often too computationally expensive, especially when the model is large. This problem has motivated a growing amount of work on approximate unlearning with and without theoretical guarantees.

In high-stakes applications, such as removing user data upon request, a mere attempt at unlearning is insufficient; a verifiable guarantee of removal is required. However, many proposed unlearning algorithms do not offer a certifiable guarantee of removal. Instead, they rely on empirical evaluations to demonstrate sufficient removal. On one hand, rigorous evaluation methods have been developed (Triantafillou et al., 2024; Hayes et al., 2025), but they are so computationally intensive that the pipeline of running an unlearning algorithm and then verifying it is often more expensive than RFS itself. On the other hand, cheaper empirical metrics have proven to be misleading. Not only do they systematically overestimate the effectiveness of unlearning algorithms (Hayes et al., 2025),

but recent work has uncovered a catastrophic failure mode: knowledge thought to be forgotten can spontaneously resurface during subsequent fine-tuning, a flaw not present in models retrained from scratch (Siddiqui et al., 2025). The profound inadequacy of empirical validation, being either impractical or unreliable, motivates a critical need for *certified unlearning*, which provides provable guarantees on the removal of data by design while maintaining the computational cost and utility benefits of approximate unlearning.

Certifiable unlearning algorithms aim to provide a formal guarantee, often expressed as an indistinguishability certificate (Sekhari et al., 2021; Chien et al., 2024; Zhang et al., 2025; Guo et al., 2020) analogous to that of differential privacy (Dwork et al., 2006). The contemporaneous work of Al Mahmud et al. (2025) proposes a recipe for certified unlearning: (1) train a model $\theta$ on $D$ with additive noise and gradient clipping (Chien et al., 2024), such as in differentially private SGD (or DP-SGD), (2) fine-tune $\theta$ on $D$ to regain the performance lost due to noise and clipping, and deploy the fine-tuned model for use, and (3) when an unlearning request is made to forget $D_F$, discard the deployed model, re-fine-tune $\theta$ on $D_R$, and deploy the resulting model. Another recent work uses output perturbation and noisy fine-tuning instead of noisy pretraining to obtain a certificate (Koloskova et al., 2025). In both cases, the amount of noise needed to satisfy theoretical guarantees induces a drop in utility, but the authors claim that they yield more favorable utility vs. compute cost tradeoffs compared to RFS at the expense of weakening the unlearning guarantee.

In this paper, we conduct a systematic evaluation of these claims. To provide the strongest possible chance for noise-based methods to succeed, we (1) optimize the above-mentioned methods, and (2) focus on a relaxed goal of *sample-level unlearning*, a weaker but still meaningful guarantee compared to the *forget-set-level unlearning* studied in prior work (Sekhari et al., 2021; Chien et al., 2024; Guo et al., 2020). This weaker requirement allows for models to be trained with substantially less noise. We then fine-tune these models on the retain set without noise to maximize utility recovery. This setup is designed to create the most favorable conditions for noise-based methods to outperform RFS.

Our central finding is that, even for this weaker notion of certified unlearning, current methods fail to provide a desirable tradeoff between cost, utility, and indistinguishability in practical deep learning settings. While certified methods may offer theoretical advantages in specific, often convex or well-behaved, regimes, our experiments show that in the training paradigms most relevant to modern deep learning, the simple baseline of RFS is often computationally efficient that these complex methods provide no practical benefit. Specifically, we show the shortcomings of certifiable unlearning in both of the dominant machine learning paradigms: training models from scratch and fine-tuning strong, publicly available checkpoints. Through extensive experiments on both vision and large language models (LLMs), we show that simple retraining consistently offers better tradeoff curves compared to methods based on noisy training. We identify two primary failure modes (FM) in our evaluation (detailed in Section 3) that explain this phenomenon. In brief, the failure modes we uncover are:

**FM 1:** When starting from a random initialization, noisy training methods find suboptimal solutions that are difficult for later post-processing to escape.

**FM 2:** When fine-tuning a strong pretrained model, retraining on the retain set is already so efficient that methods with additional computational overhead from noisy training provide no practical benefit.

Our results demonstrate that achieving certified unlearning with favorable tradeoffs remains a widely open problem, as the prevailing noise-based methods do not appear competitive against the powerful RFS baseline in realistic settings, and we call for the community to investigate new definitions and alternative techniques. Beyond the immediate context of unlearning, our findings provide critical insights for the broader differential privacy community and practitioners using noisy optimization. Specifically, our work serves as a case study on the adaptability of privately trained models, suggesting that the suboptimal solutions found by noisy training can significantly hinder a model's ability to adapt to subsequent data distribution shifts.

## 2 BACKGROUND

### 2.1 DIFFERENTIAL PRIVACY

Differential privacy (Dwork et al., 2006) is a mathematical definition of privacy that bounds the influence that any single individual in the training data has on the output of the model. Specifically,

an algorithm satisfies differential privacy (DP) if for any two datasets that differ on one individual's training data, the probability of seeing any set of potential models is roughly the same regardless of which dataset was used in training. Formally, we say that a randomized algorithm $A(\cdot)$ is $(\varepsilon, \delta)$-differentially private if for every pair of datasets $D$ and $D'$ differing on at most one training example and every set of outputs $S$

$$\Pr[A(D) \in S] \leq e^{\varepsilon} \Pr[A(D') \in S] + \delta.$$

In machine learning, the de facto algorithm used to learn models with privacy guarantees is differentially private SGD (Abadi et al., 2016) or DP-SGD. This learning algorithm is a variant of noisy SGD that first applies per-sample gradient clipping to limit the influence of any given example in the batch, then adds noise, calibrated to the clipping norm, to the average of the clipped gradients in order to preserve indistinguishability. Reducing the scale of and adding noise to each model update induces slow convergence and lower overall utility compared to non-private learning algorithms. Furthermore, per-sample clipping is computationally expensive when implemented in standard deep learning frameworks, making implementations of DP-SGD roughly 3x - 10x more expensive when compared to standard SGD (Beltran et al., 2024).

## 2.2 Unlearning via Differential Privacy

For a forget set $D_F \subset D$ and a model $\theta$ obtained by running a (potentially randomized) algorithm $A(\cdot)$ on $D$, the goal of unlearning is to design a (potentially randomized) algorithm $U(\cdot)$ that: (1) removes the effect of $D_F$ from $\theta$, (2) maintains the utility of $\theta$, and (3) is computationally efficient. A straightforward solution is to simply have $U(\cdot)$ ignore $\theta$ and rerun $A(\cdot)$ on $D_R = D \setminus D_F$. This "retrain from scratch" (RFS) approach achieves perfect unlearning and utility but can be computationally prohibitive. We therefore seek unlearning methods that can achieve more favorable computational efficiency at the expense of minor loss in utility and unlearning quality.

To control the unlearning quality, we can consider the statistical similarity between $U(\theta, D_F)$ and $A(D_R)$ as done in Neel et al. (2021). However, this is restrictive as it can only be achieved by using learning algorithms $A(\cdot)$ that satisfy DP and post-processing algorithms $U(\cdot)$ that do not touch any subset of the $D$, which implies low utility. To circumvent this issue, we quantify the quality of unlearning using a weaker notion that measures statistical similarity between $U(\theta, D_F)$ and $U\big(A(D_R), \emptyset\big)$ according to the following $(\varepsilon, \delta)$-unlearning definition from Sekhari et al. (2021).

**Definition 1** (($\varepsilon, \delta$)-Unlearning)**.** For *all* datasets D and deletion requests (or forget sets) $D_F \subseteq D$ such that $|D_F| \leq m$, and $S \subseteq \mathcal{S}$, a learning algorithm $A(\cdot)$ and an unlearning algorithm $U(\cdot)$ is $(\varepsilon, \delta)$-unlearning if

$$\Pr[U(A(D), D_F) \in S] \leq e^{\varepsilon} \Pr[U(A(D \setminus D_F), \emptyset) \in S] + \delta$$

While Sekhari et al. (2021) show that $(\varepsilon, \delta)$-DP (for datasets $D$ and $D'$ that differ on at most $m$ training samples) implies $(\varepsilon, \delta)$-unlearning (for forget sets of size $|D_F| \leq m$), their main result is a separation between the definitions of DP and unlearning in their "deletion capacity," or ability to enforce indistinguishability for many samples while maintaining high utility. Our work builds upon this implication by applying noiseless post-processing on the retain set to improve overall utility while not suffering the computational cost of per-sample gradient computation or minimum error rates to satisfy differential privacy for all samples.

## 2.3 Certifiable Unlearning

Certifiable unlearning (Bourtoule et al., 2021) aims to provide theoretical guarantees for unlearning quality, which circumvents any reliance on empirical verification which could be computationally expensive (Triantafillou et al., 2024; Hayes et al., 2025), inconsistent across forget sets (Zhao et al., 2024; Fan et al., 2024), and give a false sense of forgetting (Hayes et al., 2025; Siddiqui et al., 2025). While retraining from scratch is a certifiable unlearning method with perfect unlearning of the forget set (Bourtoule et al., 2021), the majority of methods in this space aim to yield DP-style indistinguishability guarantees in exchange for computational efficiency. While recent works provide formal guarantees for unlearning, there remains a sizable gap between their promises and practical utility. Building upon the definition from Sekhari et al. (2021), algorithms that leverage

noisy training (Chien et al., 2024; Koloskova et al., 2025) or output perturbation (Guo et al., 2020; Koloskova et al., 2025; Zhang et al., 2025) offer certificates of indistinguishability between the unlearned model and the model retrained from scratch. However, these guarantees are often non-vacuous only in regimes where retraining from scratch is already computationally feasible, such as for strongly convex objectives, or with trivial values for $\varepsilon$ and $\delta$.

## 3 OUR HYPOTHESES AND FAILURE MODES OF CERTIFIED UNLEARNING

The promise of certified unlearning (under the most widely-used definitions) is to offer meaningful certificates of unlearning while not requiring that the model owner retrains from scratch on the retain set, which would yield a perfect $(0, 0)$-unlearning guarantee. Algorithms satisfying inexact unlearning guarantees should then necessarily offer some amount of improvement in total computational cost or utility at the expense of unlearning quality when compared to retraining from scratch. However, in this work, we demonstrate that existing algorithms which satisfy indistinguishability guarantees rarely offer satisfactory enough trade-offs between these three metrics to make them worthwhile compared to retraining from scratch in practical settings.

Effectively, all of the proposed inexact, certifiable unlearning algorithms rely on additive noise Chien et al. (2024); Koloskova et al. (2025); Zhang et al. (2025), as in differentially private learning. Thus, there is an expected and necessary utility penalty incurred in order to maintain the unlearning certificate for each forget request. When implementing machine learning with sample level differential privacy, noise addition causes randomly initialized models to underfit the training data. This utility penalty is exacerbated for large models like LLMs because the total amount of noise increases in the number of trainable model parameters. Despite these challenges, noisy training methods for privacy have seen success in large models when initializing training at a strong, pretrained checkpoint. Recent works on certifiable unlearning have empirically tested their algorithms in this training paradigm, and they often report high utility with strong unlearning guarantees. We push back on these results and identify the following two failure modes of the current state of certifiable unlearning:

**FM 1:** When training a model from scratch with DP-SGD, the noisy optimizer finds a solution in a suboptimal part of the loss landscape. During the noiseless fine-tuning phase, the model either gets stuck in this "bad" loss basin (if using a sufficiently low learning rate) or must spend a significant amount of computational power to "escape" the basin and find a better solution (if using a high learning rate). Furthermore, starting from a suboptimal, noisy model induces slow convergence, which yields a marginal utility gain and renders the additional compute spent on preprocessing pointless.

**FM 2:** Fine-tuning from strong, publicly available initializations has become a primary training paradigm, as it is infeasible for most entities to train state-of-the-art models from scratch. In this setting, "retraining from scratch" means fine-tuning a pretrained model on the retain set. This process is already computationally cheap and provides comparable utility to starting from a noisy fine-tuned initialization.

### 3.1 CERTIFIED UNLEARNING FROM DIFFERENTIAL PRIVACY

In this work, we study an unlearning algorithm that uses the minimal amount of noisy training necessary to yield meaningful certificates of unlearning for samples in the forget set. The two-phase unlearning method we study is shown in Algorithm 1. First, a model is trained on the full dataset with DP guarantees, typically using DP-SGD, to create a single checkpoint with an unlearning certificate. Then, for each unlearning request, the algorithm fine-tunes a copy of this checkpoint on the retain set *without* noise in an attempt to recover utility.

A variant of this algorithm was recently studied by Al Mahmud et al. (2025), who also found that noiseless fine-tuning after a DP pre-training phase can achieve high final utility. However, their analysis focuses solely on the utility of the final converged models and overlooks the crucial dimension of computational cost. This presents an incomplete picture, since in the fine-tuning paradigm, different model initialization trained on the same data may converge to the same solution Frankle et al. (2020). As we show in Figure 1c and Figure 2a, while both Algorithm 1 and RFS can reach the same final utility, focusing only on this aspect is misleading. Analyzing the cost *and* utility over the course of

---

**Algorithm 1** Certifiable Unlearning with Sample-Level Guarantees

---

1: **Input:** Full dataset $D$, initial model $\theta_0$, noise multiplier $\sigma$, fine-tuning epochs $T$.

    **Preprocessing (One-time Setup):**
2: $\theta_{DP} \leftarrow \texttt{DP-SGD}(D, \theta_0, \sigma)$            ▷ Pretrain model with DP guarantees
3: $D_F \leftarrow \emptyset$                                   ▷ Initialize the forget set
4: $\theta_{curr} \leftarrow \texttt{UNLEARN}(\theta_{DP}, D, \emptyset, T)$      ▷ Recover utility and publish initial model

5: **function** UNLEARN($\theta_{DP}, D, D_F, T$)           ▷ Run for each unlearning request
6:      $D_R \leftarrow D \setminus D_F$                          ▷ Create new retain set
7:      $\theta_{curr} \leftarrow \texttt{fine-tune}(\theta_{DP}, D_R, T)$      ▷ Noiseless fine_tuning on $D_R$
8:      **return** $\theta_{curr}$                           ▷ Publish unlearned model

---

the unlearning phase reveals that RFS converges so rapidly in this setting that it consistently offers better trade-offs.

## 4 EVALUATION

We provide results from empirical testing of our hypotheses across multiple datasets and modalities. We show that our simple unlearning algorithm, despite offering significantly better utility and weaker unlearning guarantees compared to state-of-the-art approaches, still does not clearly improve upon retraining from scratch. In all of our experiments, we average the results of runs with five random seeds and perform extensive sweeps over choices of learning rate, optimizer, learning rate scheduler, $L_2$ regularization, forget set size $|D_F|$, noiseless post-processing epochs $T$, and noise multiplier $\sigma$. In order to make the fairest comparisons possible, we tune the hyperparameters which explicitly varied in each figure (e.g. learning rate) to maximize the unlearning algorithm's utility across runs. More details on the choices of hyperparameters used in this section can be found in Appendix B.1.

### 4.1 FM 1. CERTIFIABLE UNLEARNING WITH A RANDOMLY INITIALIZED MODEL

To investigate the first failure mode, we begin by training randomly initialized models with privacy guarantees using DP-SGD (Abadi et al., 2016). Due to the high-dimensional, non-convex loss landscape and the isotropic noise added by DP-SGD, the models consistently converge to suboptimal solutions (Ganesh et al., 2023) when compared to noiseless training. We find that despite offering a better initialization than a random initialization in terms of validation accuracy, starting from a noisy checkpoint to satisfy $(\varepsilon, \delta)$-unlearning does not offer desirable trade-offs when compared with exact unlearning.

We train randomly initialized ResNet-18 (He et al., 2015) models on the CIFAR-10 (Krizhevsky, 2009) dataset to test the first proposed failure mode. In this set of experiments, we first preprocess the models by training with DP-SGD using the Opacus (Yousefpour et al., 2021) library. To understand how different levels of unlearning quality, or indistinguishability, impact the utility of the final, unlearned model, we run the same experiment for increasing values of the DP noise multiplier $\sigma \in \{0.25, 0.5, 1.0\}$ for 20 total epochs, saving model checkpoints at the end of each epoch. We report the validation accuracy of the unlearned models over $T = 5$ noiseless fine-tuning epochs, starting from the DP model.

As shown in Figure 1a, despite having higher validation accuracy from the start (mostly due to the fact that the DP model has seen the data while the RFS model has not), retraining from scratch quickly matches, then overtakes the DP checkpoint's utility. The slow convergence of the DP checkpoint over the noiseless epochs is due to the model being underfit and trapped in the suboptimal loss basin found during the initial noisy training phase, an observation made in prior work on private optimization (Ganesh et al., 2023). We find this creates a problem for the noiseless optimizer: using a larger learning rate causes instability that forces the model out of the local minimum, while a low learning rate is too weak to escape. As a consequence, the certifiable model begins to saturate at a lower validation accuracy than the unconstrained, RFS baseline.

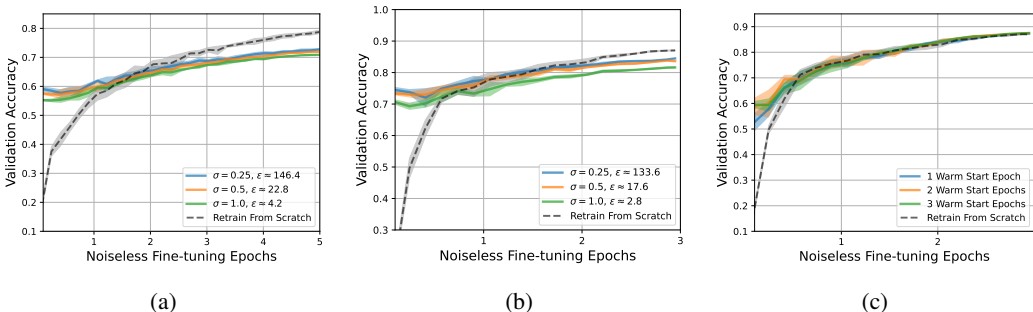

(a)             (b)             (c)

Figure 1: Comparison of utility between Algorithm 1 and RFS across three types of model initializations. The figures are ordered by increasing advantage for our algorithm from left to right, from the most challenging setting to the most favorable: (a) a random initialization , (b) a public, pretrained checkpoint , and (c) a "warm start" checkpoint. All three subfigures correspond to CIFAR-10 models which were pretrained for 10 DP epochs before unlearning 1% of $D$ ($|D_F| = 500$). A grid containing the figures for different amounts of DP pretraining and experiment details can be found in Appendix B.2.

## 4.2 FM 2. CERTIFIABLE UNLEARNING WITH A PRETRAINED MODEL

We explore the second failure mode of certified unlearning, by starting with public, pretrained initializations, rather than random weights as in Section 4.1. Prior works have shown that instantiating DP-SGD with a strong, pretrained checkpoint can yield error rates comparable to non-private learning (Yu et al., 2022; Ganesh et al., 2023). This is because the initial phase of learning generalizable representations from particular training samples has already been completed, which allows the noisy, private learning algorithm to focus only on adaptation to the new task. While this situation is ideal for private learning, as the model doesn't need to heavily depend on the sensitive fine-tuning data, it also means that retraining from scratch becomes significantly cheaper. Our evaluation shows that, while both Algorithm 1 and RFS yield unlearned models with comparable utility, RFS only takes a fraction of a single epoch to do so while also providing better unlearning quality.

### 4.2.1 VISION

We fine-tune ResNet-18 (He et al., 2015) models, pretrained on ImageNet Deng et al. (2009), on CIFAR-10 (Krizhevsky, 2009) to test the second proposed failure mode in the vision setting. We again preprocess the models by creating checkpoints that are prepared for certified unlearning. However, this time we fine-tune the pretrained models with DP-SGD for a total of 10 epochs. To measure trade-offs between unlearning quality, and utility for different computational costs, we again sweep over increasing values of the DP noise multiplier $\sigma \in \{0.25, 0.5, 1.0\}$, saving model checkpoints at the end of each epoch. We report the validation accuracy of the unlearned models over $T = 3$ noiseless fine-tuning epochs, starting from each of the noisy checkpoints.

Figure 1b shows that retraining from the ImageNet checkpoint rapidly matches the DP checkpoint's utility in less than a single noiseless fine-tuning epoch. Despite starting from a strong, pretrained checkpoint noisy pretraining offers little to no benefit as the model trained from "scratch" (i.e., from the pretrained checkpoint). Even in the case where the DP checkpoint is pretrained for 10 epochs, the head-start pretraining provides is so short-lived that it offers no practical advantage over RFS. The RFS baseline achieves a comparable validation accuracy in just half a fine-tuning epoch, which renders the 10 DP epochs a sunk computational cost.

### 4.2.2 LANGUAGE

To test the second failure mode in the language setting, we fine-tune pre-trained Pythia-1B (Biderman et al., 2023) models on the Alpaca instruction-tuning dataset (Taori et al., 2023). As in the vision experiments, we preprocess the models by creating noisy checkpoints prepared for certified unlearning. We do this by fine-tuning the pretrained Pythia models with DP-AdamW for a total of 3 epochs using the `dp-transformers` library (Wutschitz et al., 2022). We sweep over increasing values

of the noise multiplier $\sigma \in \{0.1, \dots, 1.0\}$, saving a model checkpoint for each noise scale. We report the performance of the unlearned models on a held-out test set after a single ($T = 1$) noiseless post-processing epoch, starting from each of the DP checkpoints.

Consistent with our findings on CIFAR-10, our language model experiments show that retraining from "scratch" offers competitive cost, utility, and unlearning quality trade-offs. As shown in Figure 2a, the utility advantage provided by the three epochs of noisy pre-training is minimal; the RFS baseline converges to a comparable validation perplexity within the first 10% of a single noiseless fine-tuning epoch. This rapid convergence of RFS reinforces the impracticality of the upfront cost of DP training. Foundation models like Pythia 1B are specifically trained to be generalizable to many downstream tasks. This means that the vast majority of learning has already been done on public data, so the model requires very little adaptation to the (sensitive) fine-tuning data to achieve low perplexity. This inherent efficiency makes the RFS baseline extremely fast when compared to our current methods.

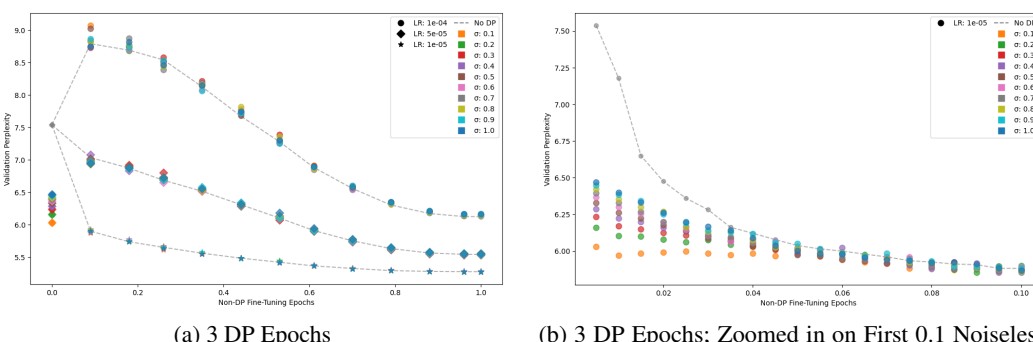

(a) 3 DP Epochs

(b) 3 DP Epochs; Zoomed in on First 0.1 Noiseless Epochs

Figure 2: Utility of Algorithm 1 vs. retraining from scratch (No DP) when instruction-tuning Pythia 1B on the Alpaca dataset

### 4.2.3 AIDING CONVERGENCE WITH WARM STARTS

We observe that the language models in Section 4.2.2 converge to the same final perplexity (Figure 2), consistent with prior work on fine-tuning dynamics (Frankle et al., 2020), while our vision models do not (Figure 1b). We hypothesize this is because fine-tuning the ResNet-18 requires replacing its final layer (which has 1000 prediction classes for ImageNet) with a randomly initialized one (which has 10 prediction classes for CIFAR-10), introducing a source of randomness not present when fine-tuning Pythia 1B for a generative task over the same vocabulary as in pretraining.

To verify this, we ran an additional experiment where we first "warm-start" the pretrained ResNet-18 by first training it with a low learning rate for one to three epochs on a held-out subset of CIFAR-10 with 5k samples. Using this new, warm-start model as the initial checkpoint for DP training with $\sigma = 1.0$ eliminates the final performance gap with the RFS baseline, as shown in Figure 1c. This result confirms that the randomly initialized layer was the cause of the inconsistent convergence and aligns the vision results with our findings on Pythia-1B. We note that the initial accuracy of the "warm-started" DP checkpoints in Figure 1c is lower than their counterparts in Figure 1b. This is a consequence of our hyperparameter tuning; a smaller learning rate was required during DP fine-tuning to preserve the weights learned during the warm-start phase, as the higher learning rate would negate its benefit. Despite this lower starting accuracy, the warm start on auxiliary samples from CIFAR-10 enables the noisy models to successfully converge to the same final accuracy as the RFS baseline.

### 4.3 PLOTTING TRADE-OFFS

Using the data from our evaluation of **FM 2**, we plot the trade-offs between compute, utility, and unlearning quality to visualize how the upfront cost of noisy pretraining amortizes over multiple unlearning requests. We define the total computational cost after the $k^{\text{th}}$ forget request as follows:

$$T_{\text{total}} = \frac{c \times T_{DP}}{k} + T_{UL} \times k$$

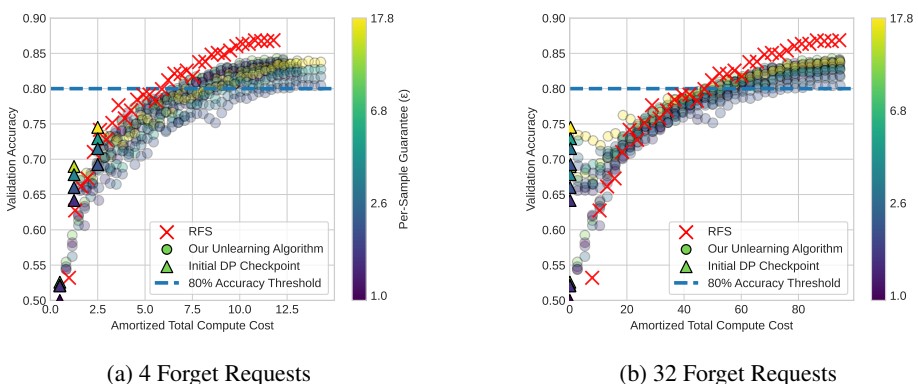

(a) 4 Forget Requests                    (b) 32 Forget Requests

Figure 3: Visualization of the achievable trade-off regions of Algorithm 1 and the RFS baseline over several forget requests. Each point is a (Cost, Accuracy, Unlearning Quality) triple.

Here, the initial pretraining cost of $T_{DP}$ DP epochs (scaled by a factor $c$ that represents the ratio between the cost of one DP epoch and one noiseless epoch) is amortized over $k$ requests, while the fine-tuning cost of ($T_{UL}$ epochs per request, accumulates linearly in the $k$ forget requests. While our hardware configuration yields a cost ratio of $c \approx 5$ when using the Opacus Yousefpour et al. (2021) implementation of DP-SGD, we adopt an optimistic $c = 1$ for our analysis. This choice reflects recent work which shows that DP-SGD, with proper configurations and optimizations like ghost clipping (Li et al., 2022), can achieve near-identical compute costs compared to standard training (McKenna et al., 2025).

Figure 3 shows the resulting trade-off curves. Each point represents a ($T_{\text{total}}$, accuracy, $\varepsilon$) triple. The triples corresponding to RFS are marked with a red **X** and have a cost of $T_{\text{total}} = T_{UL} \times k$ with unlearning quality $\varepsilon = 0$. For Algorithm 1, triples from initial checkpoints (i.e. when $T_{UL} = 0$) are marked with triangles ($\triangle$), while triples taken during fine-tuning are marked with circles ($\circ$). Although each of the $k$ requests composes to a forget-set-level guarantee of ($50k\epsilon$, $k\delta$), the color bar reports per-sample $\varepsilon$ guarantee. We include an 80% accuracy threshold (dashed blue line) to denote a reasonable target utility level. The plots reveal two key findings: (1) For a small number of forget requests, RFS dominates Algorithm 1, and (2) for larger values of $k$, the trade-offs of Algorithm 1 improve over RFS, but only in the region well below our target accuracy threshold of 80%, making the marginal utility benefit of DP pretraining futile given the additional compute cost.

### 4.4 COMPARISON WITH EXISTING METHODS

We compare our approach to two recent certified unlearning algorithms to show that the method we study throughout this work is indeed offering desirable trade-offs compared to existing work. Model clipping, proposed by Koloskova et al. (2025), is based on output perturbation. Their method first trains a standard, noiseless model, then applies clipping and a perturbation to its weights to yield a certifiable checkpoint. Similar to our algorithm, they then use a noiseless fine-tuning phase on the retain set to recover utility. In contrast, Chien et al. (2024) propose Langevin unlearning, based on continuous noisy updates during unlearning. This approach yields a stronger, forget-set-level guarantee that degrades more slowly in the size of the forget set. However, the stronger guarantee makes it difficult to provide non-vacuous guarantees in non-convex settings which are the norm in deep learning.

To compare with the algorithm proposed by Koloskova et al. (2025), we reproduce their results and run Algorithm 1 on the same, small (20k parameters) convolutional neural network used in their experiments. At each total compute cost (as determined by the sum of DP and noiseless epochs where the number of DP epochs < the number of noiseless epochs), for all methods, we report the best validation accuracy over all combinations of pretraining and fine-tuning epochs summing up to the target compute budget. Figure 4a shows that both Algorithm 1 and the unlearning algorithm proposed by Koloskova et al. (2025) are able to achieve better utility for less computation when compared with RFS. However, we note that: (1) the initial clipped and perturbed model offers a marginal benefit for small amounts of compute, (2) our method achieves comparable, or better, validation accuracy, and

(3) our experimental setup uses models with over $500\times$ the number of parameters compared to the models considered in their evaluation. Given that the error of noisy training grows in the dimension of the model, we believe the phenomena we observe become evident as model size increases.

We also run Algorithm 1 using the experimental setup from (Chien et al., 2024). In their experiments, they train a linear model on CIFAR10 features from a ResNet model pretrained on ImageNet and unlearn a single sample using their proposed Langevin unlearning algorithm. Since Chien et al. (2024) report accuracy after unlearning a single sample, we can make a fair comparison between the two guarantees. For comparison, we run our DP pretraining for a single epoch and apply noiseless fine-tuning for a single epoch. We benchmark our algorithm against the publicly available implementation of Langevin unlearning and find that our method is superior both in computational cost and utility. On identical hardware, we observe that Langevin unlearning is prohibitively slow, requiring roughly $100\times$ more computation time for a single unlearning request. As shown in Figure 4b, for the same level of certifiability, Algorithm 1 consistently achieves significantly higher utility.

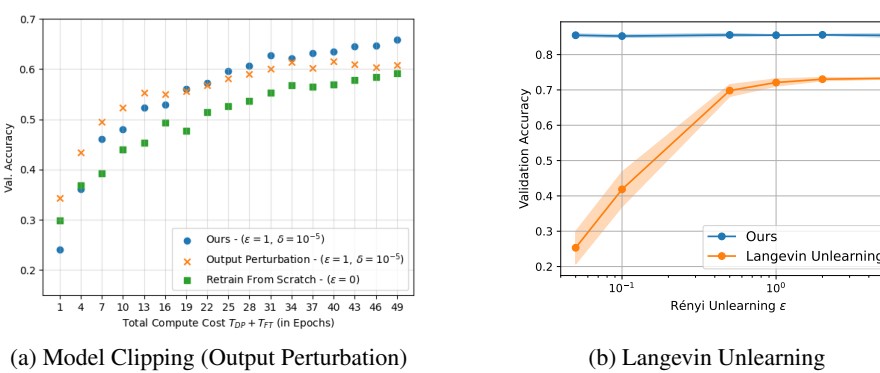

(a) Model Clipping (Output Perturbation)  (b) Langevin Unlearning

Figure 4: Comparison between Algorithm 1, model clipping (Koloskova et al., 2025), and Langevin unlearning (Chien et al., 2024), when unlearning a single sample.

## 5 DISCUSSION AND FUTURE WORK

In this work, we conduct a critical evaluation of certified unlearning and find that the simple baseline of RFS consistently offers favourable trade-offs compared to current DP-based methods. We uncover two fundamental failure modes. When training with DP from a random initialization, noisy optimizers find suboptimal solutions that are difficult to recover from during post-processing. In the now-dominant fine-tuning paradigm where we instead start from a pretrained model, the RFS baseline is already so computationally efficient that the overhead of DP-based methods provides no practical benefit, rendering them a sunk computational cost. Our findings reveal a fundamental mismatch between claims made in the certifiable unlearning literature and the ability of DP-unlearning-based methods to decisively beat simple baselines in practice.

Based on this, we make the following recommendations: First, the community should continue exploring alternative definitions of unlearning that do not rely on strict, DP-style indistinguishability. As we have shown, the cost of this worst-case guarantee can often be prohibitive. Second, new algorithms are needed that better exploit the known separation between our current notions of certified unlearning and DP. For instance, a promising direction is to design mechanisms that only enforce indistinguishability for the forget set, rather than globally degrading the entire model during a computationally expensive, noisy training phase. The primary challenge in designing these algorithms lies in the ability to offer such guarantees while also minimizing total compute cost. While methods like the algorithm proposed by Koloskova et al. (2025) represent a step in this direction, future work must show whether these kinds of approaches can be scaled to larger models. Finally, a fundamental theoretical question remains that the community should aim to answer: *What is the minimum utility penalty one must pay to satisfy current definitions of unlearning in the regime where it is more efficient than RFS?* While these kinds of fundamental limitations are well-understood for DP Bun et al. (2018); Dwork et al. (2015), establishing these barriers for unlearning is a necessary next step to guide the search for practical and optimal algorithms.

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

## A  ADDITIONAL BACKGROUND

In this section, we discuss prior work on machine unlearning not covered in the main text.

### A.1  UNCERTIFIED UNLEARNING

In contrast, uncertified unlearning methods do not attempt to provide a formal guarantee for the degree to which the effect of the forget set is removed from the model. As such, these methods are liberated from having to use gradient clipping, noisy training or any other mechanism that is required for facilitating theoretical guarantees. This freedom opens up a broad design space for building creative unlearning algorithms, and indeed a plethora of diverse methods have been proposed. We describe some notable methods below.

"fine-tuning" simply fine-tunes the model on the retain set, attempting to rely on catastrophic forgetting to erase the effect of the forget set. Jia et al. (2023) enhance this approach with sparsity-promoting objectives, further incentivizing forgetting. "NegGrad" (Golatkar et al., 2020) takes a more active approach, fine-tuning the model in the negative direction of the gradient on the forget set. To address the damage caused to the model's utility from pure NegGrad, Kurmanji et al. (2023) propose to simultaneously perform gradient descent on the forget set, in the improved NegGrad+ variation. Kurmanji et al. (2023) also proposed SCRUB, a teacher-student method that selectively (on the forget set) "disagrees" with the all-knowing original ("teacher") model, to scrub the influence of the forget set while maintaining performance on the retain set. "Random label" (Graves et al., 2021)

randomly relabels the forget set, and then fine-tunes on it, disrupting the associations the model had originally learned about the forget set examples. Golatkar et al. (2020) propose a procedure that adds noise to destroy weights that are informative for the forget set but not the retain set, using estimates of second-order gradients. A recent line of work propose to perform unlearning on only a (cleverly-selected) subset of the network's parameters, chosen via estimates of saliency (Fan et al., 2023; Foster et al., 2024) or estimates of where memorization of the given forget set examples has occurred (Torkzadehmahani et al., 2024).

However, as discussed earlier, these methods necessitate careful rigorous evaluation to obtain empirical guarantees about their unlearning quality. Because current state-of-the-art evaluation protocols are prohibitively expensive, uncertified unlearning methods may not be appropriate for applications where guarantees are required.

## B  EVALUATION DETAILS AND ADDITIONAL FIGURES

Here, we provide additional details about the experiments discussed in Section 4

### B.1  HYPERPARAMETERS

Across all experiments, we performed hyperparameter sweeps to find settings that maximized the final validation accuracy for both our unlearning algorithm and the retrain from scratch (RFS) baseline. The final hyperparameters used to generate the figures in the main body are detailed in the tables below. All vision models were trained on a single NVIDIA 4070 12GB GPU and language models were trained on 2 - 4 NVIDIA A100 GPUs.

Table 1: Hyperparameters for vision experiments with a randomly initialized ResNet-18 trained on CIFAR-10 (Figure 1a).

| Parameter | DP Pretraining | Unlearning |
| --- | --- | --- |
| Model | ResNet-18 | ResNet-18 |
| Optimizer | SGD | AdamW |
| Learning Rate | $1 \times 10^{-1}$ | $1 \times 10^{-3}$ |
| Scheduler | — | OneCycleLR |
| Batch Size | 2048 | 128 |
| Weight Decay | — | $1 \times 10^{-5}$ |
| Momentum | 0.9 | — |
| Warmup % | — | 10% |
| Max Grad Norm (DP) | 3.0 | — |
| Epochs | 20 | 5 |

Table 2: Hyperparameters for vision experiments with a pretrained ResNet-18 fine-tuned on CIFAR-10 (Figure 2).

| Parameter | DP Fine-tuning | Unlearning |
| --- | --- | --- |
| Model | ResNet-18 (ImageNet) | ResNet-18 (ImageNet) |
| Optimizer | SGD | AdamW |
| Learning Rate | $10^{-1}$ w/o Warm Start $\mid$ $10^{-2}$ w/ Warm Start | $5 \times 10^{-4}$ |
| Scheduler | — | OneCycleLr |
| Batch Size | 1024 | 128 |
| Weight Decay | — | $1 \times 10^{-5}$ |
| Momentum | 0.9 | — |
| Warmup % | — | 10% |
| Max Grad Norm (DP) | 1.0 | — |
| Epochs | 10 | 3 |

Table 3: Hyperparameters for RFS with a randomly initialized ResNet-18 trained on CIFAR-10 (Figure 1a).

| Parameter | RFS |
|---|---|
| Model | ResNet-18 (ImageNet) |
| Optimizer | AdamW |
| Learning Rate | $1 \times 10^{-3}$ |
| Scheduler | OneCycleLr |
| Batch Size | 128 |
| Weight Decay | $1 \times 10^{-2}$ |
| Warmup % | 40% |
| Epochs | 5 |

Table 4: Hyperparameters for RFS with a pretrained ResNet-18 fine-tuned on CIFAR-10 (Figures 1b) and 1c.

| Parameter | RFS |
|---|---|
| Model | ResNet-18 (ImageNet) |
| Optimizer | AdamW |
| Learning Rate | $5 \times 10^{-4}$ |
| Scheduler | OneCycleLr |
| Batch Size | 128 |
| Weight Decay | $1 \times 10^{-3}$ |
| Warmup % | 30% |
| Epochs | 3 |

Table 5: Hyperparameters for language experiments with Pythia-1B fine-tuned on Alpaca (Figure 2a).

| Parameter | DP Fine-tuning | Unlearning / RFS |
|---|---|---|
| Model | Pythia-1B | Pythia-1B |
| Optimizer | DP-AdamW | AdamW |
| Learning Rate | $2 \times 10^{-4}$ | $1 \times 10^{-5}$ |
| Scheduler | LinearWarmup | LinearWarmup |
| Batch Size | 64 | 16 |
| Weight Decay | 0.01 | 0.001 |
| Warmup Ratio | 0.03 | 0.03 |
| Max Grad Norm (DP) | 1.0 | — |
| Epochs | 3 | 1 |

## B.2 ADDITIONAL FIGURES FOR CIFAR-10 EXPERIMENTS

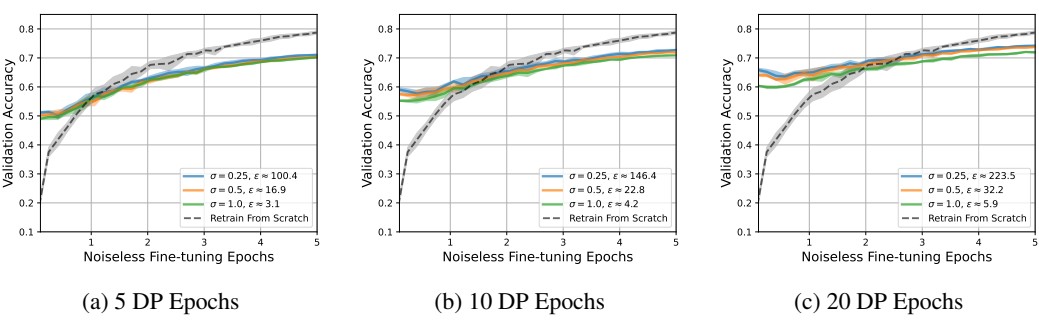

**Row 1:** Utility of Algorithm 1 vs. retraining from scratch when starting from a randomly initialized ResNet-18 on CIFAR-10 when forgetting 1% of $D$ ($|D_F| = 500$).

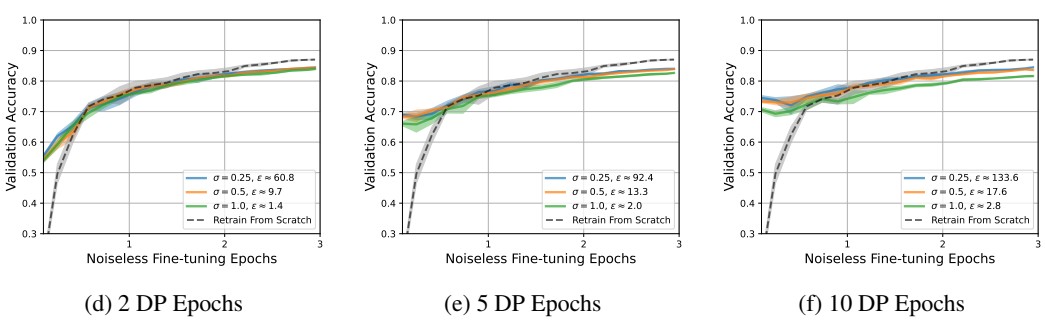

**Row 2:** Utility of Algorithm 1 vs. retraining from scratch when starting from a pretrained ResNet-18 on CIFAR-10; forgetting 1% of $D$ ($|D_F| = 500$).

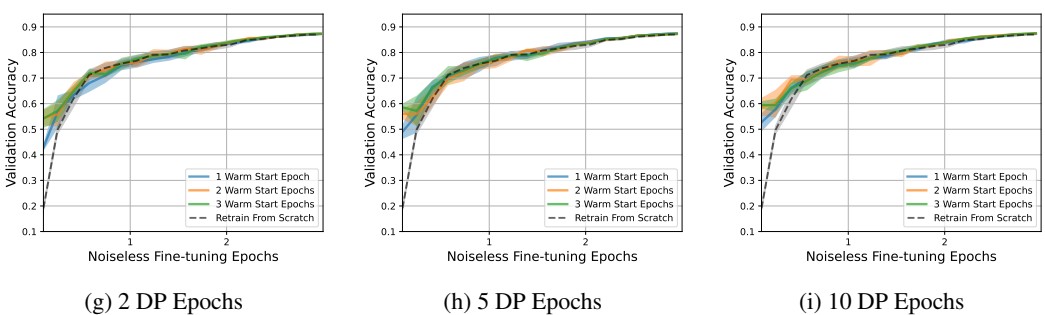

**Row 3:** Effect on model utility after unlearning 1% of $D$ ($|D_F| = 500$) with noise multiplier $\sigma = 1.0$ (or per-sample $\varepsilon = \{2.1, 3.1, 4.2\}$ for (a), (b), and (c), respectively) when running a *"warm start"* on an auxiliary subset of CIFAR-10 before DP pretraining

Figure 5: Comparison of utility between Algorithm 1 and RFS across three types of model initializations. The rows are ordered by increasing advantage for our algorithm, from the most challenging setting to the most favorable: random initialization (Row 1), a public, pretrained checkpoint (Row 2), and a "warm start" checkpoint (Row 3).

## C USE OF LARGE LANGUAGE MODELS

Large language models were used to aid in writing by helping to condense long paragraphs, rephrasing sentences, and checking for typos.

