# OpenReview forum: "Should We Forget About Certified Unlearning? Evaluating the Pitfalls of Noisy Methods"
_ICLR.cc/2026/Conference — Submitted to ICLR 2026_

### Official Review · Reviewer_bmNL · 2025-10-29

**Soundness:** 2
**Presentation:** 2
**Contribution:** 2
**Rating:** 4
**Confidence:** 4

**Summary:**

This paper provides a critical evaluation of certified unlearning methods, focusing specifically on approaches based on differentially private (DP) training. The authors conduct experiments in vision and language settings to compare these DP-unlearning techniques against the baseline of retraining from scratch (RFS). The paper's central claim is a negative result: it argues that current DP-based methods fail to offer a compelling tradeoff, often suffering from poor utility (due to DP noise guiding models to suboptimal solutions) or failing to provide a significant computational benefit over a simple RFS baseline, especially when starting from a public pretrained model.

**Strengths:**

1. The paper addresses a very timely and critical question for the field. As unlearning moves from theory to practice, we need rigorous evaluations to understand if current methods are actually viable.

2. It's good to see the evaluation extend beyond vision tasks to include larger language models. This is a crucial domain for unlearning.

**Weaknesses:**

I've ordered these in terms of importance, from high-level conceptual issues to more specific experimental concerns.

1. **Overly Broad Claim:** The paper's main conclusion that "certified unlearning" as a concept is not "worthwhile" seems too strong and not fully supported by the experiments. The evaluation focuses on one specific family of methods (DP-SGD based). It largely ignores other established classes of certified unlearning, such as sharding-based approaches (e.g., SISA), which have different trade-offs. The conclusion should be more precisely narrowed to the methods actually tested.

2. **Potentially Premature Conclusion:** The paper's strong negative claim risks being premature. Certified unlearning for complex, non-convex tasks is a very new research area, with rigorous theoretical guarantees only just emerging. It's highly likely that the initial DP-based methods evaluated here are just a first step. The paper's conclusion seems to over-generalize from the limitations of these early-stage methods, rather than framing them as a baseline for much-needed future practical progress.

3. **Experimental Scenarios Don't Test the Premise:** The core problem unlearning aims to solve is avoiding expensive retraining. However, the experiments seem to focus on relatively easy settings, such as forgetting a single sample. In this scenario, RFS (or simple re-finetuning from a public model) is expected to be a very strong and cheap baseline. The paper's claims would be much more impactful if they tested scenarios where RFS is genuinely prohibitive, such as forgetting larger, non-IID fractions of the dataset, as explored in other works.

4. **Incomplete Comparison to Prior Work:** The comparison to Koloskova et al. 2025 feels incomplete. It's not clear why the authors didn't compare against stronger methods from that paper, such as the variant called "gradient clipping" in the aforementioned paper, which was claimed to work better. The "model clipping" method used here seems to be the simpler "output perturbation" baseline, which is known to be weaker, and may not correspond to the method named "model clipping" in Koloskova et al. 2025's paper. To strengthen the negative claim, it's important to show that even the best versions of these DP-based methods are insufficient.

5. **Limited Novelty of Core Insight:** The finding that DP-SGD can lead to suboptimal solutions and trade-offs in utility is a known (though important) fact in the DP literature. Once again, given that certified unlearning for non-convex tasks is a very new field, finding that the first-generation methods are not yet practical is a useful, but perhaps not surprising, result.

6. **Limited Language Model Metrics:** While including language tasks is a strength, the evaluation is limited to perplexity. This doesn't give a full picture of model quality. It would be nice to see other standard metrics (e.g., ROUGE for summarization, or accuracy on downstream tasks) to understand the practical impact on utility.

**Questions:**

1. Could the authors please provide an explanation for Figure 2.b? It seems quite odd that all perplexity curves converge early to the same point. What might be causing this behavior?

2. Could you elaborate on the choice to focus on the DP-SGD method (Algorithm 1) also used in Koloskova et al. 2025, rather than the noiseless pretraining setting they also studied?

3. Following on that, why were other methods from that paper, like so-called "gradient clipping", not included in the main comparisons (Figures 1-3)? It would be helpful to understand if they were tested and performed similarly, or if they were omitted for other reasons.

4. Do you believe your negative claims hold for all certified unlearning paradigms, including non-DP-based methods like SISA, or should the paper's conclusion be more tightly scoped to DP-based approaches?

---

### Official Review · Reviewer_B46p · 2025-11-01

**Soundness:** 4
**Presentation:** 4
**Contribution:** 3
**Rating:** 4
**Confidence:** 4

**Summary:**

In this paper, the authors critically evaluate differential privacy (DP)–based certified unlearning methods that aim to remove specific data influences without retraining from scratch. They test these approaches across vision and language tasks and find that, contrary to prior claims, DP-unlearning offers no clear advantage in efficiency or performance compared to full retraining. The study identifies two main issues: DP training from random initialization leads to poor model quality, while starting from pretrained models makes unlearning unnecessary since simple fine-tuning achieves better results faster. Overall, the work provides an important negative result, questioning the practicality of DP-based certified unlearning.

**Strengths:**

1. Well-written and clearly presented: The paper is well-organized and easy to follow, with clear motivation, experimental design, and conclusions that make a complex topic accessible.

2. Highlights an overlooked aspect: It draws attention to an often-overlooked issue—the actual practicality and trade-offs of differential privacy–based certified unlearning in real-world scenarios.

3. Valuable negative result: The authors provide a rare but important negative finding, showing through systematic experiments that current DP-unlearning methods fail to outperform retraining. This insight is valuable for guiding future research toward more effective unlearning approaches.

**Weaknesses:**

1. The title — "Should We Forget About Certified Unlearning? Evaluating the Pitfalls of Noisy Methods" — feels overly broad and somewhat misleading. The paper only examines one specific class of certified unlearning methods, namely those based on DP-SGD training followed by fine-tuning on the retain set, considering just two scenarios: training from scratch and fine-tuning a pretrained model. However, other classes of certified or exact unlearning methods exist—such as those using Newton-based updates with additive noise—which are not explored here. In addition, approaches like [a] and [b] that handle the case without any retain data, focusing solely on the forget set, represent fundamentally different settings where retraining is infeasible. Therefore, the claim implied by the title that we should “forget” the entire field of certified unlearning appears too strong given the limited scope of evaluation.

[a] Fast Yet Effective Machine Unlearning (Tarun et. al)
[b] Towards Source-Free Machine Unlearning (Ahmed et. al)

2. The empirical evaluation is limited to relatively small-scale datasets such as CIFAR-10, which makes it difficult to generalize the conclusions to large-scale scenarios where retraining from scratch is practically infeasible—such as with current large language models trained on massive data corpora. Hence, the claim that DP-based certified unlearning is not worthwhile may not hold in realistic, large-data contexts.

3. Furthermore, the analysis overlooks the fact that existing DP-SGD–based certified unlearning methods are inherently approximate and rely on relaxed assumptions. While these limitations are well known, dismissing the entire line of research risks discouraging progress. A more balanced conclusion would acknowledge that, although current methods are imperfect, continued efforts to reduce assumptions and bridge the gap between theory and practice are essential to advancing the field.

**Questions:**

While the paper includes experiments on the Pythia-1B language model, this still represents a relatively moderate scale compared to current large foundation models. How do the authors expect their conclusions to hold for much larger models—such as multi-billion parameter LLMs—where retraining from scratch is prohibitively expensive and DP-based certified unlearning might still offer practical advantages despite its theoretical and empirical limitations?

Overall, the effort is commendable, and more such rigorous “reality-check” studies are needed to objectively assess assumptions and guide the unlearning community in the right direction.

---

### Official Review · Reviewer_W8qh · 2025-11-01

**Soundness:** 2
**Presentation:** 2
**Contribution:** 2
**Rating:** 2
**Confidence:** 5

**Summary:**

This paper examines certified unlearning methods that rely on differentially private (DP) mechanisms (e.g., DP-SGD) to provide formal guarantees that models have forgotten specific data points. The authors perform an extensive empirical study across vision (CIFAR-10, ResNet-18) and language (Pythia-1B, Alpaca) tasks to test whether these methods actually outperform simple retraining from scratch (RFS). They claim that there are two key "failure modes" of DP methods:

FM1: Noisy DP training leads to suboptimal local minima that hinder subsequent fine-tuning and yield inferior utility.

FM2: Retraining (fine-tuning from a pretrained checkpoint) is efficient enough.

**Strengths:**

1. The paper is clearly written and organized, with well-labeled figures and sections, though at times it overexplains basic concepts such as DP-SGD.

2. The authors perform several experiments on both image data and text data.

3. The topic is timely given current interest in unlearning and DP.

**Weaknesses:**

1. The conclusions extend far beyond what the experiments can justify. Results on ResNet-18/CIFAR-10 and Pythia-1B/Alpaca cannot represent the behavior of larger or domain-specific systems where **retraining is expensive**. The authors ignore contexts where certified unlearning is actually needed (e.g., regulated deployments, multi-tenant models). Thus, the sweeping claim that “certified unlearning is not worthwhile” is unsupported by the limited evidence.

2. FM1—noisy DP training leads to suboptimal local minima—is not promising and is already well-known. However, this can be mitigated by a more fine-grained hyper-parameter tuning. The main reason for this claim in the paper is that their hyperparameters are not “optimal.” This is also evident from Figure 1, where the accuracy of fine-tuning a publicly pretrained model on CIFAR-10 is only 80%. However, DP fine-tuning on CIFAR-10 typically achieves an accuracy above 90% (cf. https://arxiv.org/abs/2204.13650). Overall, the experiments are not convincing.


3. The study treats DP noise as an intrinsic flaw rather than a "hyperparameter" and does not explore modern variants of DP-SGD (e.g., adaptive clipping, DP-LoRA, or tighter privacy accountant) that can dramatically reduce noise. These methods should have been applied in the experiments. This is another evidence that the experimental results are not convincing.

4.  Unlearning effectiveness is not actually tested in this paper. No membership inference attacks are conducted. All conclusions rely solely on DP certificates and accuracy metrics, neither of which confirm that the influence of the forget set has been removed. Moreover, the forget set size (|DF| = 1 % of CIFAR-10) is arbitrary and unexplored—no scaling experiments on forget-set size.

5. The key claim relies on an ambiguous notion of “not worthwhile.” However, what thresholds of cost, privacy budget, or accuracy make a method “not worthwhile”? In regulated environments, any provable guarantee—regardless of cost—may be necessary, meaning that “worthwhile” is inherently policy-dependent. Their framing also overlooks the regulatory and compliance context that motivates certified unlearning in the first place.

Overall, the paper is structured to confirm its predetermined conclusion (“certified unlearning is not worthwhile”) rather than to test it objectively. All experiments are designed to maximize RFS’s apparent advantage—using small forget sets, simple architectures, and low training costs. The DP setup is deliberately weakened and then declared ineffective. This amounts to a self-fulfilling negative result rather than an unbiased evaluation.

**Questions:**

see the weaknesses

---

### Official Review · Reviewer_fria · 2025-11-01

**Soundness:** 2
**Presentation:** 3
**Contribution:** 3
**Rating:** 4
**Confidence:** 4

**Summary:**

This paper presents a systematic empirical evaluation of certified (DP-based) unlearning versus the simple “retrain-from-scratch” (RFS) baseline on vision and language tasks. Under realistic, large-model regimes the authors find that current noise-injection schemes consistently lose to RFS in cost–utility–guarantee trade-offs, identifying two failure modes: (i) random-init DP pre-training lands in poor loss basins that are hard to escape, and (ii) pretrained-init makes RFS so cheap that the upfront DP cost becomes sunk overhead. Recommendations to rethink definitions and explore non-DP guarantees are well-motivated.

**Strengths:**

1. The work provides the first reality check on whether certified unlearning actually beats the trivial retrain-from-scratch baseline. The authors recommend revisiting definitions and exploring alternative techniques.
2. The paper is overall well-structured.

**Weaknesses:**

1. he reviewer disagrees with the statement in FM2: "In this setting, 'retraining from scratch' means fine-tuning a pretrained model on the retain set." Regardless of the setting, fine-tuning is always a distinct unlearning method separate from retraining from scratch (RFS). Since fine-tuning-based methods themselves lack theoretical guarantees, we could also disregard the constraints required for certified unlearning and directly apply their algorithms for unlearning.
2. The paper lacks discussion on online unlearning (i.e., handling continuous unlearning requests).
3. The paper's survey of certified unlearning is insufficient, lacking discussion of recent methods such as that in [1].


   [1] Qiao, X., Zhang, M., Tang, M., & Wei, E. (2025). Hessian-Free Online Certified Unlearning. *International Conference on Learning Representations*.

**Questions:**

I would appreciate the authors’ responses to the weaknesses outlined above.

---

### Meta-Review · Area_Chair_mNvo · 2026-01-07

**Summary:**

This paper presents an empirical study of DP-based certified unlearning methods, assessing whether they are over retraining-from-scratch. The results suggest that these methods do not consistently outperform retraining.

Among four reviewers, they consistently raise concerns that the paper’s conclusions are overly broad relative to the narrow class of methods and experiments: the experiments focus on settings where retraining or re-finetuning is already cheap, and the failure cases are likely due to known limitations or hyperparameter choices, rather than fundamental deficiency of the method. Its insights are viewed as insufficient to support the strong negative claims.

There is no rebuttal provided.

**Reviewer Concerns:**

N/A
There is no rebuttal provided.

**Reviewer Scores:**

N/A
There is no rebuttal provided.

---

### Decision · Program_Chairs · 2026-01-26

Reject